# Loneliness, Wellbeing, and Social Activity in Scottish Older Adults Resulting from Social Distancing during the COVID-19 Pandemic

**DOI:** 10.3390/ijerph18094517

**Published:** 2021-04-24

**Authors:** Simone A. Tomaz, Pete Coffee, Gemma C. Ryde, Bridgitte Swales, Kacey C. Neely, Jenni Connelly, Andrew Kirkland, Louise McCabe, Karen Watchman, Federico Andreis, Jack G. Martin, Ilaria Pina, Anna C. Whittaker

**Affiliations:** 1Faculty of Health Sciences and Sport, University of Stirling, Stirling FK9 4LA, UK; simone.tomaz@glasgow.ac.uk (S.A.T.); peter.coffee@stir.ac.uk (P.C.); gemma.ryde@stir.ac.uk (G.C.R.); bridgitte.swales@stir.ac.uk (B.S.); kacey.neely@stir.ac.uk (K.C.N.); jenni.connelly1@stir.ac.uk (J.C.); andrew.kirkland@stir.ac.uk (A.K.); louise.mccabe@stir.ac.uk (L.M.); karen.watchman@stir.ac.uk (K.W.); j.g.martin@stir.ac.uk (J.G.M.); ilaria.pina@stir.ac.uk (I.P.); 2College of Medical, Veterinary and Life Sciences, University of Glasgow, Glasgow G1 1BQ, UK; 3Department of Mathematics and Statistics, Lancaster University, Lancaster LA1 4YR, UK; federico.andreis@gmail.com

**Keywords:** loneliness, social support, social isolation, social network, wellbeing

## Abstract

This study examined the impact of social distancing during the COVID-19 pandemic on loneliness, wellbeing, and social activity, including social support, in Scottish older adults. A mixed methods online survey was used to examine these factors during social distancing mid-lockdown, July 2020. Participants were asked to state whether loneliness, wellbeing, social activity, and social support had changed since pre-social distancing, and to provide details of strategies used to keep socially active. A total of 1429 adults (84% aged 60+ years) living in Scotland took part. The majority reported that social distancing regulations made them experience more loneliness and less social contact and support. Loneliness during lockdown was higher than reported norms for this age group before the pandemic. A larger social network, more social contact, and better perceived social support seemed to be protective against loneliness and poor wellbeing. Positive coping strategies reported included increasing online social contact with both existing social networks and reconnecting with previous networks, as well as increasing contact with neighbours and people in the community. This underlines the importance of addressing loneliness and social support in older adults but particularly during situations where risk of isolation is high.

## 1. Introduction

Coronavirus, or severe acute respiratory syndrome coronavirus 2 (SARS-CoV-2) that causes the coronavirus disease, COVID-19, was declared a pandemic by the World Health Organization on 11 March 2020 [1]. By 9 August 2020, it was indicated that globally, there were over 19 million confirmed cases of COVID-19 and over 700,000 deaths [2]. In the UK, equivalent data at around the same time indicated that there were 309,767 confirmed cases and 46,566 fatalities [2]. Consequently, early public health advice centred around social distancing, self-isolation in the presence of symptoms or exposure, and reducing contact with others as ways to protect against becoming infected [3]. At its most strict time point, the UK Government referred to this as a ‘lockdown’, which was primarily a directive to stay at home unless for essential purposes including buying food and exercise. The most at-risk groups (those aged 70+ and with pre-existing health conditions) were also directed to ‘shield’, which meant they were advised not to leave the house at all. The purpose of this study is to focus specifically on the impact of social distancing on psychosocial variables (loneliness, wellbeing, social contact, and social isolation), but other key important or confounding variables including sociodemographics and health behaviours were taken into account. The term ‘social distancing’ is used within this paper to reflect the terminology first used by the Scottish Government during the pandemic but refers to the need to physically distance from people.

Cross-sectional and longitudinal studies have demonstrated the negative impacts of loneliness, social isolation, and poor social support on core factors of human health from mental health to mortality [4,5,6]. This is a key issue for many older adults already, but with the current COVID-19 crisis and social distancing rules, the magnitude of the problem and its consequences for many vulnerable groups is likely to increase. As an example, social connections and social support have been shown to predict positive mental and physical health, on average contributing to the variance in health of a magnitude four times greater than the variance predicted by financial security [7]. The health and lifestyle outcomes associated with social support and isolation are varied. In older adults, social isolation has been associated with poor nutritional intake [8], poor social networks have been associated with higher experiences of stress [9], and social support, particularly from close family members, has been shown to be important for leisure time physical activity [10]. It is currently concerning that the disruption in social contact and/or low-quality social relationships could lead to feelings of social isolation and loneliness, especially in older adults [11].

A substantial volume of evidence across different cultures in the last four decades suggests that the size and quality of social networks and higher levels of social support can have a positive effect on reducing loneliness in older adults [12,13,14,15,16,17]. This protective effect of social support against the negative impact of loneliness in older adults [18,19] has been demonstrated to subsequently result in improved quality of life [20], wellbeing [21], and life satisfaction [22]. For example, close friends within social networks can reduce loneliness [15] and enhance the quality of the social network. Feelings of loneliness are directly related to satisfaction with one’s social network and neighbourhood attachment [14]. The provider of support within one’s social network is also important. This is because emotional support, particularly from family members, is negatively associated with loneliness among older people [18]. The quality of perceived support has effects on wellbeing and loneliness, with higher levels of support shown to both alleviate loneliness and improve wellbeing in older adults. This suggests that support may improve wellbeing through reducing loneliness [21]. For example, higher perceived social support related to decreased experiences of loneliness amongst South Korean older adults, and the negative effect of low social support on poor quality of life was mediated by loneliness [20]. Providing as well as receiving social support can also be beneficial; both receiving social support from and providing social support to others was found to be negatively related to loneliness in Nepalese older adults [12]. Further, among older adults who provide and receive high amounts of support, this has proven to be a protective factor against loneliness [19]. Taken together, research has shown that helping older adults to increase or maintain the size and/or quality of their social networks leads to positive effects on social support together with reducing levels of loneliness [17].

Older adults can be particularly vulnerable due to the increased risk of co-morbidities associated with ageing such as frailty, hearing or sight impairment [23]. This can make engaging in social and/or physical activity a challenge. Currently, only 53% of 65–74 year-olds and 31% of those aged 75+ years in Scotland [24] adhere to the government guidelines on weekly physical activity, showing lower levels of physical activity at older ages [25]. These low levels of physical activity are concerning because physical activity has benefits including reduced mortality and physical/mental morbidity [26]. Further, public health initiatives have used physical activity to reduce loneliness among older adults with some success [4,27]. The advice on social distancing has restricted both social and physical activity, and this may have a greater impact on groups most vulnerable to COVID-19. This includes those with health conditions and individuals >70 years who are already at greater risk of loneliness and its negative correlates due to changes in health, relationships, mobility, and finances [28]. Previous research suggests that these individuals already report high loneliness; 100,000 older people in Scotland reported feeling lonely all or most of the time, and 200,000 will go half a week without a visit or call from anyone [29]. 

Despite these pre-COVID-19 findings, research conducted in the Scotland during COVID-19 in adults (median age 53 years; [30]) has shown that older adults are doing better in terms of anxiety and depression in comparison to younger adults. This is somewhat consistent with other UK-wide studies, where the proportion of older participants experiencing mental distress is lower than that of younger participants [31,32]. These findings may indicate that for older adults, the contrast between pre-COVID-19 and current-COVID-19 times may be less pronounced. Thus, older adults may also report some positive consequences of current COVID-19 restrictions. Individuals may discover alternative ways to maintain social contact and physical activity, such as through technology, which could be used and shared to benefit others both during and beyond the current pandemic.

The aim of the present analysis was to examine the impact of social distancing during the pandemic on loneliness and wellbeing. Specific objectives were to: (1) describe levels of loneliness, wellbeing, social contact/activity, and social support during social distancing; (2) examine perceived changes in loneliness, social contact, social support, and wellbeing from before to during social distancing; (3) explore associations between psychosocial variables (loneliness, social contact, social support, and wellbeing) and other important variables such as physical activity, during social distancing; (4) explore whether levels of these psychosocial variables vary by important sociodemographics such as age, sex, deprivation, and rurality; and (5) describe strategies used by older adults to maintain socially activity during social distancing. No formal hypotheses about the direction and magnitude of perceived changes in psychosocial factors were formulated, given the lack of prior information, although it was expected that social distancing could have a negative impact on the loneliness, social contact, social support, and wellbeing of some groups of older people. 

## 2. Materials and Methods

### 2.1. Design

This cross-sectional study used a mixed methods online survey approach. Data were collected from Scottish older adults. Ethical approval for this study was obtained from the University of Stirling General University Ethics Panel (GUEP 905), and informed consent was given on the first page of the online survey or orally by telephone for the phone version. All participants provided consent for their participation. This study adheres to the guidelines described in the Declaration of Helsinki Ethical Principles for Medical Research Involving Human Subjects [33]. The resulting quantitative dataset will be available from the University of Stirling online STORRE database upon publication.

### 2.2. Participants and Procedure

The study focused mainly on older adults (defined in this study as 60+ years as per United Nations [34]) as a known at-risk group but also included younger groups vulnerable to loneliness specifically, including carers, those advised to shield, and those with a learning disability (data on this younger group will be reported elsewhere). Participants were recruited across Scotland via social media (Twitter, Facebook), emails to colleagues at other Scottish Universities, partner organisations and contacts including care homes, churches, sports clubs, charities for older adults, third sector and intermediary organisations for individuals with learning disabilities and carers, national dementia organisations, and snowballing through personal contacts. The survey went live online on the morning of 28 May 2020. Later this same day, the Scottish First Minister had announced the move from lockdown to ‘Phase 1’, which was the first stage of a proposed route map out of the toughest restrictions. The survey closed on 31 July 2020. On 30 July, the Scottish Government announced that shielding would be paused from 1 August. The details of the phases and the relevance to recruitment in this study are shown in Figure 1. Sampling was non-probabilistic; from 27 June, targeted Facebook advertising was used centring around seven major cities in Scotland and within a 40 km radius of each and focusing on those aged 40–65+ years who might see it and be able to pass it on to an older relative or friend. The non-probabilistic nature of the sample was accounted for in the analysis by carrying out checks of balance with respect to known characteristics of the target population.

To reach as wide a pool of participants as possible, the participant information sheet and consent form were also available in an easy-read format. This incorporated the use of images to support the meaning of text. Whilst this format was to support the inclusion of people with a learning disability or visual impairment, it was recognised that easy read is often preferred by most participants. It may also be preferred for participants who are not fluent in English or who have coordination difficulties for whom large print may be preferred. The pictures used on the easy-read consent form and information sheet were sourced under license from Photosymbols [35]. The information provided was concise, written in plain language and designed to be as legible as possible by using font size 14, and was provided alongside the standard recruitment materials. The option to undertake the survey by telephone further supported recruitment from wider groups, all of which is consistent with the UK government guidance on accessible communication formats [36].

### 2.3. Measures

The survey included 49 questions under the six following headings: (1) Your personal details (including sociodemographics); (2) your health and sleep (including questions about chronic illness); (3) your wellbeing; (4) your physical activity, sitting time, and screen time (reported briefly in this paper as required but in greater detail in Tomaz et al., in preparation); (5) your social contact and social activities; and (6) the effect of the pandemic on you (including a question about COVID-19 infection). Five of the forty-nine questions were open-ended questions. These were included (under the appropriate headings listed above) to explore any strategies adopted to maintain both physical (reported elsewhere) and social activity and to capture additional contextual information. The survey was designed to take no longer than 30 min to complete. Participants could ask someone to help. For those who did not use web-based technology, the option of completing it by telephone was offered.

#### 2.3.1. Sociodemographics

Sociodemographics included age, sex, ethnicity, relationship status, education, income bracket, current or previous employment status, postcode (zip code), number of people in the home, number of people requiring care within the home, and a question about whether the participant was a care provider. Where a participant responded with ‘Prefer not to say’ for any of these variables, the data were marked as ‘missing’. Income was across eight categories from <£2500 per year to over £50 k per year. For those participants that provided postcodes, urban and rural living were classified using the Scottish Government Urban Rural Classifications Breakdown [37]. Analyses used the 3-fold classification into urban (settlements of 10,000 or more people) and small town (areas with 3000–9999 people) versus rural (areas with a population of less than 3000 people), as used previously by others [38]. Quintiles of deprivation were also derived from postcodes using the Scottish Index of Multiple Deprivation (SIMD), where 1 equates to being the most deprived and 5 least deprived [39].

#### 2.3.2. Health and Health Behaviours

Health was assessed using standardised questions about common medical diagnoses, and medication taken for those diagnoses. Health behaviours assessed were sleep, walking, physical activity category regarding meeting government guidelines (full physical activity data are presented elsewhere), and screen time. Participants reported on their typical bedtime, waking up time, and actual hours of sleep (time in bed). The questions in the survey were asked in a ‘dropdown’ menu in 30 min intervals (bedtime and waking up) and in hour-long intervals for actual hours of sleep. Participants were able to answer ‘other’ if there was not a time appropriate for them to report. The answers for the bedtime and waking up questions were converted to decimals so that an approximate average for both bedtime and waking up could be reported (e.g., 8 p.m. = 20.0 = 20h00; 7:30 a.m. = 7.5 = 07h30). The answers for time in bed were used to place participants into categories according to the National Sleep Foundation sleep time recommendations [40] for those aged under 65 years, and those aged 65+ years, which split participants into five categories: short sleepers; short but may be appropriate sleepers; ideal sleepers; long but may be appropriate sleepers; and long sleepers. For use as an explanatory variable in the present analyses, these were combined to form three categories: short sleepers, ideal, and long sleepers. Participants were also asked whether, compared to before social distancing, they were sleeping less well, the same, or better.

Using the short International Physical Activity Questionnaire (IPAQ-S) [41], participants indicated how much time they spent walking, in moderate intensity physical activity, and in vigorous intensity physical activity, on average per day over the past seven days. Participants’ physical activity levels were then categorised as either low active, moderately active, highly active in accordance with the IPAQ manual, and also according to UK government guidelines for physical activity [42].

Screen time in the past week was assessed using similar wording to how physical activity was assessed via the IPAQ: Participants reported screen time engagement as an average time per day over the past seven days. In the absence of clear guidance on how best to clean screen time data, data for participants that reported in excess of 16 h of daily screen time (n = 1) were truncated to 16 h (960 min). It is important to note that the survey did not distinguish a type of screen time or context of screen time (e.g., sedentary screen time, type of device, passive screen use such that the person may have been watching TV while doing housework such as preparing food). For PA and screen time variables, participants were asked whether they were less, the same, or more compared to before social distancing.

#### 2.3.3. Loneliness, Wellbeing, and Social Activity including Social Support

The revised brief form of the UCLA Loneliness scale, the ULS-6, was used to measure loneliness and was chosen over longer versions to minimise participant burden. It has been validated in older adults and shows high internal consistency [43]. Internal consistency reliability in the present study was 0.83.

Wellbeing was measured using the EQ5D-3L [44], which assesses mobility, self-care, usual activities, pain/discomfort, and anxiety/depression on three-point scales to indicate level of problems or difficulty for each. It also includes a 0–100 rating of general state of health. This was scored according to the EQ5D-3L handbook [44]. Internal consistency reliability for this scale in the present study was adequate at 0.68.

Several aspects of social activity were measured. To determine social contact, participants were asked about the number of days they engaged in social activities (in the last 7 days) and average time spent in social activity (in minutes). The wording was similar to the method of assessment of physical activity over the past week using the IPAQ-S [41] for consistency of question style and in the absence of a brief measure of this type. Where participants provided a range of time in a particular activity (e.g., 15–20 min of social activity per day), the median of the range was determined and used in analysis. Where the answer was unclear, e.g., ‘?’ or ‘varies’ or ‘several hours’, this was marked as missing data (for both the time indicated and number of days of social activity reported). Both structural (social network size) and functional (perceived support, received support, provision of support) aspects of social support were explored. We estimated current reported social network size and perceived availability of support, and perceived changes in all social support variables. Social network size was estimated using the first question from the Medical Outcomes Survey Social Support Scale [45] which asks about number of close friends and relatives that one can feel at ease with and talk about what is on their mind. Where participants indicated a range, the lowest number was imputed. Where the answer was unclear, e.g., ‘many’, this was marked as missing data (n = 23). Perceived social support was measured using a brief perceived social support questionnaire (BPSSQ), a 6-item scale [46] that has been validated across several cultures [47]. Items were assessed on a 5-point Likert scale ranging from 1 (not true at all) to 5 (very true). Higher scores indicate higher levels of perceived social support. The Cronbach’s alpha in the present study was 0.88.

For loneliness and all social activity variables, including social support, participants were asked to indicate the change as ‘less’, ‘the same’, or ‘more’ since before social distancing began, e.g., “How does this compare to how you felt before social distancing started? I feel less/the same/more supported now”. With regard to wellbeing, participants were specifically asked whether their anxiety/depression had changed, as well as whether their state of health was worse, the same or better than before social distancing. Changes in phrasing, i.e., from less to less well or worse, or more to better, were used so questions made grammatical sense.

#### 2.3.4. Strategies to Maintain Social Contact and Additional Contextual Factors of the Lived Experience of during Social Distancing

The survey included five open-ended questions, for which the responses of two are reported in this manuscript. The data from the other three questions are presented elsewhere (Tomaz et al., in preparation), as they were targeted to specific changes in physical activity, and this is beyond the scope of this paper. Question 1 aimed to explore strategies older adults engaged in during social distancing to maintain social contact, whilst question 2 allowed for participants to provide any additional contextual information they felt might be relevant to the study and provide explanation for the quantitative findings. These questions were as follows: (1) *What, if any, activities or behaviours are you now engaging in to make sure you have social contact during the social distancing? Your social contact may include chatting (more often) to your neighbours, video-calling friends and family, reconnecting with old friends or attending church online. We have provided some examples of some types of social contact activities in the image below. Please feel free to be as descriptive as you like*; and (2) *Is there anything else that you can think of about social engagement, loneliness, wellbeing and/or physical activity during the pandemic that we haven’t covered in this survey? You may be as descriptive as you want.*

### 2.4. Data Analysis

First, the survey dataset was downloaded from the online host (JISC) and pre-processed in Excel for cleaning and normalisation purposes, then moved to SPSS v26 and R 4.0.2 [48] for statistical analyses. For this, descriptive statistics were computed for all variables; inferential statistics were carried out based on the 60+ age group that was found to be well-balanced with respect to some key population characteristics. Next, correlations were initially used to explore associations between questionnaire scores. Finally, a bivariate Clayton copula model [49] with Gamma margins described by generalised additive models [50] was employed to explore the association of loneliness (UCLA) and wellbeing (EQ5D-3L) with the explanatory variables jointly. The two outcomes were observed to be moderately associated (the estimated copula dependence parameter translated in a ~0.22 Kendall’s tau). Information on age, deprivation (using SIMD quintiles), rurality (3-fold definition), sex (binary), social support (BPSSQ), size of the social network, number of social hours per week, perceived health rating (on a scale of 0–100), screen time (hours per day), reported walking (minutes per week), physical activity category (low active, moderately active, highly active), and sleep category (short sleepers, ideal, and long sleepers) were included. These variables were selected on the basis of theoretical considerations and previous studies where age, deprivation, sex, social support, physical activity, and sleep were shown to be associated with loneliness and/or wellbeing. Splines [50] were used to allow for nonlinear relationships of the outcomes with numerical/ordinal explanatory variables to be described, while categorical covariates entered the model as linear terms. There were complete data for n = 976 participants.

Qualitative data in the form of open-ended questions were imported from Excel into NVIVO v12 Pro for Windows. ST and BS read through the answers to the two open-ended questions relevant to this paper. ST coded and KN and BS acted as ‘critical buddies’ during the coding process. In order to be consistent with the quantitative analysis, only qualitative data for participants aged 60 years and older are reported in this manuscript. Overall, a thematic analysis using both inductive and deductive approaches was used when analysing all the open-ended questions [51]. For question 1, we specifically wanted to explore the nature of strategies to maintain social contact. We adopted an explanatory theoretical approach [51] to the qualitative analysis of the answers to this question: the qualitative data were used to explain or substantiate quantitative findings. Another reason for this approach included the tendency for participants to answer the open-ended question regarding social activity with responses that did not provide much insight into the differences or changes in social activity (e.g., participants would answer with one or two words such as “Zoom”, “video chatting”, “FaceTime”). For question 2, the answers to this open-ended question were primarily scanned for details pertaining to earlier questions in the survey, and where appropriate, those answers were addressed (e.g., postcode input difficulties, missing options for some questions, general comments/feedback on the survey). Thereafter, the answers were read by ST and coded according to common themes that provide insight to the lived experiences of participants aged 60 years and older.

## 3. Results

### 3.1. Data Missingness and Representativeness

The survey was completed by 1444 respondents, but 15 were excluded due to not being a resident in Scotland (which was a condition of the grant funding), resulting in a sample of 1429 participants. Missingness across the dataset was low at 3.9% of the total observations spread across approximately 53% of variables. Those variables with highest missingness were ‘medication’ and ‘income’. These are explainable as only participants who had indicated that they had a chronic illness/medical condition were then asked if they were taking medication for said condition(s). Income was not obligatory as we expected that many people would not wish to disclose this [52], which would reduce the overall response rate if asked to do so. The next set of missing variables included the derived Scottish Index of Multiple Deprivation (SIMD rank, quintile) and rurality measures which were all derived from, and dependent on, participants inputting their postcode; hence, the 9% missingness is expected to be shared across them. Again, postcode was a variable we had made non-obligatory after our piloting showed that the slightest mistake in the formatting of postcode input rendered it impossible to continue with the survey. This resulted in three versions of this variable across the survey administrations. There was no convincing evidence that postcode was missing in a consistently different manner across these three versions, with relative proportions of missingness at 9.98%, 5.36%, and 9.49%. Inspection of the key outcome variables (UCLA and EQ5D-3L scores) suggested that missingness was negligible at (5.6% and 2.5%, respectively).

Representativeness was assessed through comparing the balance of the sample with respect to a set of pre-defined population variables deemed important to be adequately represented: sex, age, deprivation (as measured by SIMD quintile), and rurality (as measured by the 6-fold Urban/Rurality index). The respondents were mostly 60+ years. Explaining the imbalance, this was likely due to the way the survey advertising was formulated to target ‘older adults and those at risk of COVID-19’. For individuals over 60 years old, the definition of target population coincides with that of general Scottish population, thus allowing a direct verification of balance across the key variables above. Inspection of the age by sex distribution in the 60+-year-olds in this sample against the national survey data suggests overall comparability; similarly, for the same group, the distribution of rurality is broadly consistent with that of the reference population [53]. The sample appears, however, to be imbalanced with respect to deprivation; specifically, looking at SIMD quintiles, the most deprived group is somewhat under-represented with respect to the population, whereas we observe an over-representation of the least deprived quintile. The second to fourth quintiles appear to be fairly represented. This imbalance is probably due to both the surveying method being predominantly online and the differing non-response rates by socioeconomic status [54]. Therefore, the imbalance needs to be taken into account when interpreting the results. Due to the heterogeneity of the under-60 group, as well as information about the key variables for the reference population (individuals vulnerable to COVID-19) being very little, this paper focuses on the 60+ age group exclusively.

### 3.2. Sociodemographics

Of the 1429 participants, the majority were aged 60+ (n = 1198, 84%). Participant characteristics and sociodemographic details are shown in Table 1 for the sample. The geographic distribution of respondents across Scotland is shown in Appendix A.

The majority of respondents over the age of 60 were aged 60–69 years (n = 832, 69%); the median and interquartile range for age was 8 (63–71 years). Only four participants received the telephone interview version of the survey rather than completing it online. As expected, these individuals were older (mean age = 81.8 years). Most participants were married/cohabiting (63%), lived with one other person (56%), and were not living with someone needing theirs or external care (80%). The majority (70%) had undergraduate degree education or higher. Based on 1094 participants who provided a viable postcode, most participants were in quintiles 4 and 5 of deprivation (least deprived) (58%) and resided in urban areas (69%).

Details not shown in Table 1 include ethnicity (‘White British’ n = 1185, 99%) which reflects the Scottish population [55], employment level (‘retired’ n = 811, 68%; ‘employed/self-employed’ n = 214, 18%), and income bracket (earning < £10,000 n = 106, 11%; earning ≥ £30,000 n = 376, 38%; ‘Prefer not to say’ n = 194).

### 3.3. Health and Health Behaviours

Fifty two percent of the sample reported at least one health condition. Of those with a diagnosis of a health condition, 94% were taking prescribed medication for it. Many participants (43%) were classed as highly active and reported spending 336 min per week walking and 3.7 h per day using screens.

### 3.4. Loneliness, Wellbeing, and Social Activity Including Social Support

Participants’ mean (SD) UCLA loneliness score was 12.7, and they spent on average five days a week in social contact, which was spread across seven hours of activity time per week. Participants had a mean (SD) social network size of five people and average perceived social support rating of 3.8. On the EQ5D-3L, participants overall reported moderate wellbeing and self-reported health. The total scores as well as proportions of the sample with no, some or extreme problems under each of the five categories of wellbeing are shown in Table 2 with the other psychosocial variables. The overall level of loneliness was higher than the normative UCLA loneliness score for adults aged 60–74 years of 11.25 and the norm for those aged 75+ years of 12.14 [43]. For wellbeing and self-reported health, our participants seemed to report better mobility, self-care, and ability to perform usual activities than normative values for age group, but more problems with pain and anxiety/depression [56]. Finally, for perceived social support, scores were similar to norms reported elsewhere [46] for those aged 55–74 years (average = 4) and those over 75 years (average = 3.83).

### 3.5. Change in Loneliness, Wellbeing, and Social Activity Including Social Support

Perceived changes in participants’ loneliness, wellbeing, social activity, including social support, and physical activity since before social distancing began are shown for all variables where this question was asked for the whole sample in Table 3.

The majority reported that social distancing regulations made them experience more loneliness, social contact with fewer people, less frequent social contact, and shorter social contact time. However, perception of social support quality, amount of social support given to others, general health rating, anxiety/depression rating, and sleep volume more commonly remained the same. Screen time increased for the majority.

### 3.6. Associations between Loneliness, Wellbeing, and Social Activity Including Social Support

Loneliness was associated with a smaller social network, lower perceived social support, less social contact in the week in terms of hours per week, worse wellbeing, and poorer self-rated health. On the positive side, as expected, individuals who had a larger social network and more social contact time reported better perceived social support quality, less loneliness, and better wellbeing and health ratings. Correlations are shown in Table 4.

### 3.7. Associations with Loneliness and Wellbeing and Sociodemographic and Behavioural Variables

Figure 2 shows the interval plot of the estimated association of each categorical covariate on the outcomes’ scales (loneliness and wellbeing scores); reference categories are: ideal sleep, low PA, urban, male. Being female was observed to be associated with higher loneliness on average. The uncertainty around rurality does not allow us to rule out that living in a remote rural area (as opposed to the ‘rest of Scotland’) has no effect on loneliness score, on average, when everything else is kept at the reference level. For wellbeing, factors associated with worse wellbeing (higher score) were less than ideal sleep, low PA, and being female. Living in an accessible rural setting (as opposed to the ‘rest of Scotland’) was not found to be strongly associated with variations in wellbeing score, on average, and everything else being kept at the mean level. Being female was found to increase, on average, the wellbeing score by ~0.2 points (indicating worse wellbeing), whereas living in a remote rural area was found to reduce it by ~0.4 points on average, everything else being kept at the mean level. Appendix A contains the complete output of the statistical model.

The estimated shape of relationship of the outcomes with continuous and ordinal variables is displayed as a series of spline plots. The plots report the number of observations along the independent variable’s range (shown as a density rug at the bottom of each graph) as well as the uncertainty around the functional estimate (shown as the shaded area). Figure 3 displays the estimated relationships with loneliness (UCLA), Figure 4 with wellbeing (EQ5D-3L). Further details can be found in Appendix A.

Age, deprivation, social contact time per week, and screen time did not show marked relationships with the average loneliness (UCLA) score, whereas perceived social support and health rating appeared to be broadly inversely related to the average loneliness score. The social network size estimate has low uncertainty for social network sizes up to ~20 people and seems to relate negatively to loneliness score, such that as network size increases up to 20 people, loneliness score decreases. The lack of data points for higher sizes causes the uncertainty explosion, as evidenced by the increasingly wide bands.

Age did not show an effect on wellbeing until past 80 years of age, where evidence suggests it starts to have a detrimental effect; we note that the uncertainty around the estimate past this point is larger, due to fewer observations. Deprivation score related somewhat to lower wellbeing (indicated by lower EQ5D-3L scores) among the most deprived. Social network size is estimated to have a positive effect on wellbeing; however, uncertainty is once again large for values past ~20, due to lack of data. The effect of reported weekly social hours on wellbeing appears to be negligible. For perceived social support, there was an indication of wellbeing being worse among those with lower perceived support. Self-reported health rating exhibits a nonlinear relationship with wellbeing score. Reported low health rating scores up to ~30 (out of 100) seem to be associated with a higher average wellbeing score (i.e., worse wellbeing), to then U-turn and reverse the trend afterwards indicating that higher reported health rating is related to better wellbeing. Lastly, less screen time and greater time walking seemed to relate to better wellbeing. However, there was more uncertainty about this for values past 300 min per week walking, where the relationship flattened out to negligible and there were fewer data points, and for past 5 h of screen time where there were also fewer data points, although the negative linear relationship persisted.

### 3.8. Strategies to Maintain Social Contact

Four major themes were identified when analysing the first question (What, if any, activities or behaviours are you now engaging in to make sure you have social contact during the social distancing?). These included: (1) technology, (2) community, (3) physical activity, and (4) support and being supported. Additionally, several participants provided substantial detail of their social activity. It is worth mentioning that the answers to this question were often brief, non-grammatical, and sometimes typed as a list with little or no punctuation.

The most commonly reported anecdote from the first open-ended question was that social activity had changed, but for the most part, the change was only in how it was being conducted. Participants reported that the way they interacted with their (a) friends and family, (b) faith (through religious gatherings), (c) chosen group activities (previously face-to-face), and, to a lesser extent, (d) employer and colleagues had changed. Over 300 participants mentioned Zoom in their answer, many of whom responding to the question with “Zoom” as a one-word answer. Mentions of family gatherings and detailed catchups with immediate family were most frequently reported:


*Daily phone calls with son and daughter. Son lives a distance away and takes me for virtual walks with 2 dogs every day. As I am very familiar with the beautiful park he goes to—woodland, grassy areas, riverside, I almost feel I am there. Wider family facetime three times a week minimum. Daughter visiting at my door while socially distancing several times a week.*
—Female, 72 years, ‘rest of Scotland’

The frequency of people mentioning ‘reconnecting’ with friends and family from many years back was noted:


*During lockdown I rang old friends more frequently and reconnected with people I hadn’t spoken to for a while. It was hard not having close contact with family but we FaceTimed daily.*
—Female, 68 years, ‘rest of Scotland’

Just over one hundred and fifty participants mentioned that religious gatherings had gone online, replacing face-to-face gatherings such as Sunday Mass, prayer groups, and bible reading/study groups. Ninety-one participants reported that their social gatherings with friends and family had changed in favour of ‘games nights’. Social activities with friends and family such as bingo and quiz nights were new, and bridge was ongoing, just online. Other more ‘formal’ activities that were also mentioned frequently included book club, choir, music club, dance class (and other exercise classes such as yoga, tai chi, Pilates), and language classes. Lastly, some participants mentioned that their social activity had changed because of their work situation changing:


*Video calls with work and family. Work 2–3 times a week, family—once a week Social videos with work colleagues every week WhatsApp chat with colleagues daily—except weekends.*
—Female, 61 years, ‘accessible rural’

Community, and particularly the role of neighbours, was mentioned by over three hundred participants. Common experiences that were a direct result of lockdown included getting to know neighbours (that were unknown before) and increased interaction with both neighbours and others in the community (e.g., people at the shops or at the local park). These interactions were largely reported as positive, thus being a beneficial contributor to the social activity in this sample. Importantly, some participants mentioned that a pleasant Scottish summer (which coincided with data collection) helped with such interactions.


*I spend all day at my window and have lots of chats with friends, neighbours and innocent passers-by!*
—Female, 70 years, ‘rest of Scotland’


*The whole village is at home and out doing stuff in their gardens. We talk more than before. We talk longer than before.*
—Male, 66 years, ‘remote rural’

At least one hundred participants reported that their social activity had been linked to their physical activity. Time spent outdoors while walking for exercise, walking the dogs, or active commuting provided an opportunity for people to chat to their neighbours (as mentioned above) but also to engage with others in their community/village at common destinations (e.g., the local park). Additionally, some participants reported having ‘socially distanced’ walking meetings with friends and family, particularly once lockdown restrictions had eased:


*Since lockdown has eased I have been walking with friends, but now maintaining social distance.*
—Female, 64 years, ‘accessible rural’


*Constantly meeting & chatting with people whilst out walking—especially those whose dogs come running to me, … chatting with neighbours more.*
—Female, 64 years, ‘rest of Scotland’

Lastly, there were participants who reported that their social activity had simply just changed. They specifically mentioned that this was either through support that they were providing to others or through support that they were receiving, beyond contact with their friends and family. Reported mechanisms of support included ‘buddy systems’, online groups where people came together to support other groups including the NHS, food and essential supply deliveries to the most vulnerable, and volunteering for resilience teams and volunteering as a listener for bereavement centres. Appendix A provides additional qualitative information from participants who provided very detailed accounts of their social contact and activities.

### 3.9. Additional Contextual Factors of the Lived Experience of during the Pandemic

Seven different themes were identified in the analysis of the second question (*Is there anything else that you can think of about social engagement, loneliness, wellbeing and/or physical activity during the pandemic that we haven’t covered in this survey?*): These included: (1) frustration, (2) exhaustion, (3) feelings of being forgotten, (4) anxiety and depression, (5) sadness and grief, (6) expression of challenges, and (7) positive anecdotes. Although these themes did not always link directly to the participants’ social engagement, loneliness, wellbeing and/or physical activity per se, the responses provide a great deal of insight to the lived experience through this period of social distancing. The responses also highlight that when exploring loneliness and wellbeing, many other factors come into play.

Many participants reported their feelings of frustration, although the source of the frustration reported varied greatly. The most commonly reported sources of frustration included lockdown guidance (from the government), the media, other people, and with the NHS/other health services:


*I get frustrated that rules and regulations are so prescriptive, not allowing room for common sense. This leads to great feelings of guilt if one takes a calculated risk to care for a vulnerable relative who needs care…*
—Female, 68 years, ‘rest of Scotland’


*…Angry about being asked by my doctor [if] we wanted to be resuscitated if required, made to feel that although family and friends care society as a whole thinks we are not important in the overall scheme of things.*
—Female, 70 years, [no postcode provided]


*When I was sick I called NHS24 who advised me to call my doctor and on phoning my doctor I was advised to call NHS24. I had symptoms … The reason I was not given any attention was because I could not prove I had a fever. … I have been extremely tired since I had the virus. I phoned my doctor and the nurse said “You’ve told me all this before. You’re suffering from post viral fatigue but you’ve only had the flu”.*
—Female, 66 years, ‘rest of Scotland’


*At the start of lockdown I felt I was going to have panic attacks again. Lost faith in other human beings the way they behaved.*
—Female, 65 years, ‘rest of Scotland’

Many participants also mentioned that they were experiencing exhaustion and amotivation across several spheres of their lives, including employment:


*As a frontline professional who has had no respite since the onset of the pandemic; the expectation that I will be able to adapt to a change of working and take on considerable more responsibility is expected; regardless of my family circumstances, which are also in disarray. The support from my organization is superficial no more than ticking boxes and the expectation is great to be present for others actually, emotionally and 24/7. I feel undervalued, patronized and exhausted.*
—Female, 62 years, ‘rest of Scotland’


*As a single person living alone I was used to being on my own but did have a reasonable social life. I have missed the personal contact with people most but have found zoom meetings very helpful. I have 2 main problems—a lack of motivation to achieve anything domestically, and days when I feel very down and lonely. I tend not to contact anyone to discuss that preferring to get through the day on my own and trying to get out for some exercise which usually helps!*
—Female, 73 years, ‘rest of Scotland’

Although less commonly reported, some participants reported feelings of being forgotten. This was often linked to friends/family, medical/health conditions, and support from the government—for themselves as well as others (addressed in greater detail in the section regarding challenges for carers):


*It has been hell. My son who has Down’s syndrome, autism and epilepsy has had no support since the pandemic started, day services stopped and not knowing when they will restart is awful. I feel abandoned by the system and exhausted as I am unable to do anything for me.*
—Female, 61 years, ‘rest of Scotland’


*Not being able to work, and not qualifying for any government schemes causes anxiety. I had a part-time, self-employed job cleaning 3 times a week for 3 elderly ladies. (Because not my MAIN source income, but very important one, I don’t qualify for financial support. I am VERY angry about this) These 3 jobs gave me a lot of social contact and emotional support from the kindly [old] ladies I cleaned for. This has now gone.*
—Female, 62 years, ‘rest of Scotland’

Many participants reported feelings of anxiety, fear and uncertainty—particularly about the future:


*At first I was happy in my bubble but now I’m feeling fed up and a bit frightened as I don’t know how it’s all going to end.*
—Female, 75 years, [no postcode provided]


*There is no doubt that at times I have felt very isolated and lonely. Anxiety has a lot to play into feeling this way … not knowing when or if life will return to some normal. Not knowing when I will see close family that live over 500 miles away…*
—Female, 64 years, ‘accessible rural’

Stories of grief and sadness were also very common, and many participants expressed a longing to see loved ones. Missing major events (e.g., the birth of a grandchild or milestone birthdays) and especially missing grandchildren were mentioned frequently:


*The biggest thing for me is not being able to spend time with my grandchildren and go out with my friends. No amount of online activities can replace that.*
—Female, 66 years, ‘rest of Scotland’


*Felt loss of husband suddenly two years ago more acutely as no overnight stays from grandchildren or visits from family and friends.*
—Female, 76 years, ‘accessible rural’

Although carers did not comprise a large proportion of our sample (12%, Table 1), the challenges experienced by carers and/or those who live(d) with people shielding or with illness appeared to be unique to those mentioned above:


*I am a carer for elderly parents, one has dementia and one has cancer. I am shopping for them and their only source of support at the present time. Pre Covid they had home help, gardener and my sisters and other family members were visiting regularly. The isolation has had a huge impact on their wellbeing and consequently on mine. The time involved in supporting them has increased significantly.*
—Female, 60 years, ‘remote rural’


*We have been caring for [daughter] for 47 years. Her Care package was dramatically cut To one and a half hours per week from day one. She did not make the shielding list!! Couldn’t get shopping. Feel the learning disabled abandoned by government planning…*
—Female, 68 years, ‘accessible rural’


*I take care of my husband, who has been disabled for the past 22 years. There is no respite during the COVID-19 epidemic, so I have no free time, and feel my mental health is suffering, with no clear optimism for the future*
—Female, 71 years, ‘rest of Scotland’

Although the response to the open-ended question was largely negative, there were many participants that reported positive stories, happy anecdotes, and ‘success stories’ relating to their mental health, physical health, and their ability to cope. Particularly, many participants stated that the pandemic had made them experience a greater sense of gratitude for the things that bring them joy:


*I felt fine with social distancing and was not in any hurry to participate in any easing of lockdown but now that I have started meeting friends whilst remaining physically distanced it is absolutely wonderful. Really enjoying simple pleasures in life.*
—Female, 62 years, ‘rest of Scotland’


*… Far from being awful, this time has been like having a gift that I couldn’t have anticipated and has made me reflect on how I will live my life moving forwards to reduce stress and have a lasting impact on health and wellbeing*
—Female, 60 years, [no postcode provided]


*I am very lucky that I am in lockdown with a loving husband in beautiful surroundings with plenty of open space. Fortunately, all my family, including my 94 year old mother and 86 year old father in law, are keeping well so this pandemic has been a positive experience for me personally. I’m aware that this is not the case for many people.*
—Female, 60 years, ‘accessible rural’

## 4. Discussion

The aim of this study was to examine the impact of social distancing during the pandemic on loneliness, social and physical activity, and wellbeing. The five specific objectives were to: (1) describe levels of loneliness, wellbeing, social contact/activity, and social support during social distancing; (2) examine perceived changes in loneliness, social contact, social support, and wellbeing from before to during social distancing; (3) explore associations between psychosocial variables (loneliness, social contact, social support, and wellbeing) and other important variables such as PA during social distancing; (4) explore whether levels of these psychosocial variables vary by important sociodemographics such as age, sex, deprivation, and rurality; and (5) describe strategies used by older adults to maintain social activity during social distancing.

This study is important because it adds to knowledge regarding how people cope in periods where the risk of isolation is high. Although specific to the COVID-19 pandemic, this study has wider implications of helping us to understand the impact of social distancing and social isolation on older people. The key findings from this study showed that social distancing, overall, had a detrimental effect on loneliness and wellbeing, i.e., increasing loneliness and decreasing wellbeing. Higher than normative levels of loneliness were reported as well as somewhat lower levels of social support. A minority of participants recognised positive aspects during periods of social distancing. Most participants reported using web conferencing technology to maintain social contact. Fewer participants reported the pandemic and periods of social distancing brought additional meaning to life through highlighting what was important to them.

Four ‘take home messages’ are highlighted and discussed in greater detail in this discussion. First, the quantitative analysis showed that evidence was found of both loneliness and wellbeing being associated with sex, perceived social support, health rating, and size of social network; however, these associations were not always as expected. Second, in addition to the above, wellbeing was also found to be associated with physical activity level, time spent walking, and sleep. To a certain degree, the nature of these associations was as expected. Further, the qualitative data provided a lot of valuable contextual information about the role of physical activity within social activity. Although physical activity did not appear to be directly associated with loneliness and is also not the focus of this paper, we presented qualitative evidence that physical activity had an important role to play in remaining socially active. Third, the qualitative data provided ample information about how social activity changed for our participants. As such, in most instances, the qualitative data supported the quantitative findings (e.g., screen time increased for most participants), although sometimes, the qualitative findings were not always reflected in the quantitative results (e.g., screen time was not associated with loneliness or wellbeing), making the contextual information from the qualitative aspect of the study even more valuable in understanding this aspect. Fourth and importantly, the qualitative data provided a clear indication that all Scottish older adults did not experience the period of social distancing between March and July 2020 in the same way. This point is especially crucial when it comes to providing recommendations to what is evidently a diverse group of people in terms of their lived experience of social distancing.

With regard to the associations with loneliness: poorer perceived social support and a lower health rating were related to higher loneliness, but a larger social network (up to 20 individuals) had a positive relationship with loneliness. Although wellbeing is discussed in greater detail below, it is worth noting here that wellbeing was also associated with social network size. Based on this, it might be speculated that social distancing guidance due to COVID-19 either created or exacerbated pre-existing difficulties in engaging in social contact, which then had a negative impact on loneliness. This is supported by the majority of participants in this study reporting that social distancing regulations made them experience more loneliness but also less social contact in terms of time, frequency, and number of people they were in contact with. This is in line with findings from the Scottish Health Survey Telephone Survey [55] which reported that one in ten people who had been advised to shield felt lonely most of the time in comparison to one in twenty who were not advised to shield. Our study shows that this also appears to extend to reduced social contact, which is likely to underpin the increased loneliness. The Scottish Health Survey also found that, prior to the pandemic, 79% of over 75s had contact with someone on most days of the week [24] and an average network size of five people, similar to the findings in this study. This shows some positive outcomes from this research, as older adults who increased or maintained the size and/or quality of their social networks had better social support and lower loneliness [17]. This is somewhat in line with a previous survey where the wellbeing of older adults was less influenced by loneliness than that of younger adults, perhaps underlining the greater contrast between pre-pandemic and current pandemic social contact for younger people [30,31,32]. However, a substantial proportion here still did report high levels of loneliness despite this social contact, and social support scores were slightly lower than normative values [46], underlining the importance of addressing this issue among older adults too.

It was also interesting to note that despite increased feelings of loneliness and less social contact, close to 60% of participants reported experiencing no change in the quality of their social support experiences. Close to a quarter reported a lower quality in social support, and a minority reported an improvement in social support. A similar pattern was observed for wellbeing (discussed later). This suggests that perceived social support was low even before social distancing commenced. This is problematic because social support, and its reported positive impact on loneliness in older adults [18,19], has previously been demonstrated to improve quality of life [20], wellbeing [21], and life satisfaction [22]. This is, to some extent, in line with the present study, in that poorer perceived social support related to higher loneliness; however, perceived social support did not directly relate to wellbeing (discussed below), although a lower health rating was associated with worse loneliness. Previous research has shown that helping older adults to increase or maintain the size and/or quality of their social networks leads to positive effects on social support together with reducing levels of loneliness [17]. Given that the causal direction of associations cannot be determined by this study or previous observational studies, the present data could be interpreted as demonstrating similar links between perceived social support, social network size, loneliness, and wellbeing. Further, the size of one’s social network appears beneficial but may only be valuable up to a certain size of network as demonstrated in previous studies [57], and here, such that having a larger social network benefitted loneliness but only up to a network size of 20 individuals, beyond which we were not able to determine an association due to lack of data points. A further explanation is that there are also individual differences in how people experience and perceive loneliness, which means the relationship between social contact/networks and loneliness is not always straightforward—see, e.g., [58,59].

Another consideration in relation to loneliness is that of social strain: ineffective support relationships can also be a source of loneliness. While this was not directly assessed in this study, another study done by Chen and colleagues (2014) reported evidence that relational strains involving spouse/partner, children, family, and friends can intensify loneliness in older adults [21]. This might explain why poorer perceived social support related to higher loneliness even when 60% of the 60+ years age group did not live alone. The quality as well as the quantity of support is important. This is also clear from some of the qualitative findings (discussed in greater detail below). For some individuals, being at home with their loved ones continually due to social distancing was a source of stress, often due to the lack of respite from caring duties. Conversely, a number of participants noted their gratitude at having a loving partner or family in their home.

Despite the changes in wellbeing from before social distancing and associations described above, it is worth noting that for wellbeing, the present sample reported better than normative values (for the target age group) for the mobility, self-care, and ability to perform usual activities categories [56]. This may be due to the recruitment approach within the project and reflect that respondents self-selected and were able to engage with an online survey, which might be less likely among less independent and active older people; it may also reflect the deprivation imbalance in the sample noted earlier, such that fewer were from highly deprived categories. However, our sample also reported more problems with pain and anxiety/depression [56]. This may be attributable to the pandemic and social distancing situation. Research reports that treatment for chronic pain has been hit twice by the pandemic, first by restricted access to healthcare for those already receiving treatment for chronic pain and second by a growing number of people affected by pain as an outcome of COVID-19 [60]. Based on other research findings, the increased anxiety and depression levels could potentially be explained by the change, and in some cases loss, of social networks due to COVID-19 [61]. The qualitative findings reported in our paper suggest that at least one other source of anxiety included uncertainty around how or when the pandemic will end. Although a thorough recruitment strategy was adopted, this sample was the relatively highly educated, less deprived, and physically active compared to the Scottish population at large, Thus, finding that some had difficulties due to COVID-19 social distancing among this group suggests that these effects may be even worse for those in lower socioeconomic groups and/or who are less active. This finding is in line with that of a previous survey which showed that poorer wellbeing was reported among women and more deprived groups [30].

Less than ideal sleep and low levels of physical activity were also found to be associated with poorer wellbeing. While their relationship with sleep and physical activity will be discussed in more detail elsewhere (Tomaz et al., in preparation), it is worth discussing sleep briefly and addressing the association with physical activity in the context of social activity. Firstly, regarding the relationship between wellbeing and sleep: this finding is expected and not only supported by other studies including older adults [62,63] but also supports the rationale of the sleep time duration recommendations by the National Sleep Foundation [40]. Regarding physical activity, based on the findings reported in this study specifically, lower levels of reported physical activity are not only linked to wellbeing, but also concerning because the qualitative results clearly indicated that outdoor physical activity provided opportunities for safe social contact while being active. Beyond the pandemic, encouraging safe social contact through physical activity may be an important recommendation to reduce loneliness, increase social activity, and improve social support for Scottish older adults experiencing these issues. It may also be important for the improvement of wellbeing. This is further supported by both research studies done prior to the pandemic [64] as well as a study conducted in Canadian older adults during the strictest phase of pandemic restrictions [65]. In the Scottish context, the recommendations for improving loneliness, wellbeing, and social contact may be through encouraging older adults to get to know their neighbours better, getting involved with local buddying systems, and engaging in physical activity such as daily walks in the community.

Another element that was made clear in the qualitative results included further explanations for the lack of change in social support quality and provision of support to others remaining similar to before social distancing began. Participants who lost resources which previously helped them to cope with life, such as support services or social networks, tended to struggle more with the unpredictable nature of the pandemic. For others, learning to engage socially using technology such as Zoom, WhatsApp, or FaceTime, or who found new sources of satisfaction in life, reported a tendency of coping better. Social groups such as religious groups, clubs, and other organisations have also used technology more during COVID-19, and this has been important for a great deal of older adults. Participants reported actively looking for new social contact during restrictions, such as contacting friends who they had not spoken to in years and increasing interaction with neighbours and other members of their community. They also used strategies such as engaging with technology to maintain contact with those they would normally see on a more regular basis. This is particularly positive given that both receiving social support from and providing social support relates to decreased loneliness [12,19]. This may be a means by which participants who engaged in such activity were able to prevent even higher levels of loneliness during social distancing, and in some cases were able to experience lower levels of loneliness. Thus, educating older adults to increase their digital literacy level and use of remote social interactions could be a really important tool for addressing issues such as loneliness [66].

### 4.1. Strengths and Limitations

This study is not without limitations. First, due to its cross-sectional nature, we are unable to explore change over time or potential mediation in the relationships that were explored. However, the aim of the study was to provide timely evidence surrounding the impact of social distancing rather than how this altered over time, and we were keen not to overburden participants by asking them to complete the same measures for their current status and prior to social distancing. A strength is our considerable sample size, giving insight into these issues. Second, the present analysis focused on investigating the potential associations of loneliness and wellbeing scores with social support and other variables, but there remains the possibility of reverse causation—for example, older adults in social isolation may not have access to sufficient social support resources. Research has demonstrated that being socially isolated and lonely is associated with the most social support gaps [59]. There is also the possibility of influence by other unmeasured potential confounding variables. We have been careful not to claim causality in our results, and a strength was the inclusion of many theoretically important variables, such as sociodemographics and other health behaviours in the model. Third, due to non-probabilistic sampling, and lack of existing data on norms within those aged below 60 years, we were only able to attempt inference on the 60+ age group. However, this reflected the main target and majority of our sample, and recruitment overreached the initial target. Fourth, there were a number of limitations with the survey that are worth noting for future studies: (1) Participants were not asked whether they resided in a care home facility, which may have contributed to the variation in numbers reported for social support by a small proportion of participants. (2) The qualitative data were not as rich as would be achieved from focus groups or interviews. However, we were still able to gather considerable useful explanatory information, and the design suited the restrictions on research due to COVID-19. (3) The response rate was lower in men and people from higher deprivation ranks. This is a common finding in research data [67,68]. A strength is that the sample was deemed balanced relative to Scottish population norms for the key variables of age, sex, deprivation and rurality; hence, there was enough confidence to continue with inferential statistics. However, this is important because as previous evidence suggests, loneliness scores and PA levels may have had a proportionately greater negative effect on these younger groups [69]. Therefore, it is a priority to find ways to engage with these groups now and post-COVID-19.

### 4.2. Implications

The findings of this study suggest that overall, older adults in Scotland have experienced multiple changes over the period of the pandemic. As we emerge from the pandemic, policy and practice implications from the results are that they suggest most older adults in Scotland would benefit from guidance and support in reducing loneliness, increasing social contact, and improving overall wellbeing. In the Scottish context, the recommendations for improving loneliness, wellbeing, and social contact may be through encouraging older adults to get to know their neighbours better, getting involved with local buddying systems and community initiatives including via digital means, and engaging in physical activity such as daily walks in the community. These recommendations may be communicated by both policy makers as well as practitioners who regularly engage with older people.

Future research should examine more nuanced effects of social support upon loneliness. For example, the effects of older adults providing social support was not captured in this study but has been found to be positive in reducing providers’ experiences of loneliness [12]. In the present study, generic levels of perceived support from participants’ support providers were examined, but research has demonstrated that support from close relationships such as family ties and support from siblings is of relevance to older adults’ experiences of loneliness [18,70]. Additionally, cross-cultural effects for types of social support have been found on loneliness such that, for older adults in Spain, instrumental support was found to be a protective factor against loneliness but, in the Netherlands, emotional support was found to be a protective factor [19], suggesting that type of support as well as geographical location would be a fruitful line of enquiry. Finally, the average closeness of older adults’ social networks, rather than network size, has been found to be an important correlate for lower levels of loneliness previously [13]. In the present study, the question about network size specified about close ties that they could rely on, but it is possible that participants did not answer it in this way and simply reported their number of social contacts. Further investigation into the closeness of social support sources might reveal more information regarding the specific mechanisms by which this influences loneliness and wellbeing. It would also be valuable to examine these mechanisms within specific subgroups such as people with dementia, or carers.

## 5. Conclusions

This study examined the impact of social distancing during the pandemic and has highlighted a range of positive and negative findings regarding loneliness, wellbeing, and social activity, including social support as a result of the period of social distancing in Scottish older adults. Most older adults reported that social distancing regulations made them experience more loneliness and less social contact and support. Loneliness was noted to be higher than reported norms for this age group before the pandemic. However, a larger social network, more social contact, and better perceived social support seemed to be protective against loneliness and poor wellbeing. These findings are important because they provide lessons on how better to help and/or support older adults with respect to loneliness, wellbeing, and social activity, including social support. The results of this study may be applicable to the future, both in and outwith pandemic situations. Overall, it is clear that there was a great deal of variety in lived experience within this group of Scottish older adults during the period of social distancing between March and July 2020. Quantitative analysis of factors associated with loneliness and wellbeing showed that sex, perceived social support, health rating, and size of social network were all key correlates. Additionally, wellbeing was also associated with physical activity level, time spent walking, and sleep. This study also explored strategies used by older adults to remain socially active, and the qualitative aspect highlighted that encouraging safe social contact through physical activity and engaging with people in the community may be an important recommendation to reduce loneliness, improve wellbeing, increase social activity, and improve social support. Additionally, the use of technology and connecting with social groups through technology was also important in maintaining social contact. Future interventions may wish to focus on affording older people the opportunity to find sources of satisfaction and meaning in life through quality opportunities to interact socially.

## Figures and Tables

**Figure 1 ijerph-18-04517-f001:**
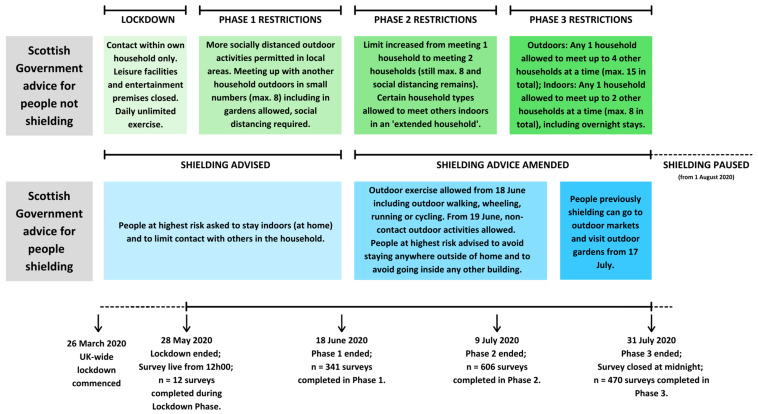
Phases of Scotland’s 2020 lockdown; advice comparison for persons not shielding versus persons shielding in relation to data collection for this study.

**Figure 2 ijerph-18-04517-f002:**
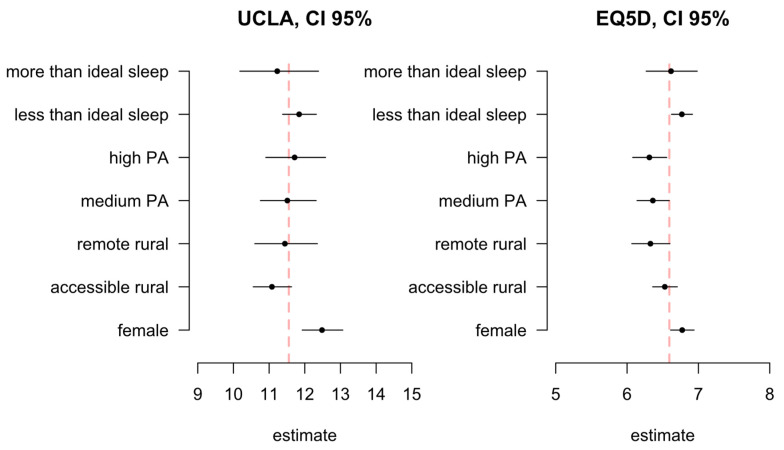
Associations with loneliness (UCLA, **left**) and wellbeing (EQ5D-3L, **right**) for categorical variables included in the model: sleep quantity, physical activity level, rurality, and gender. The dashed line indicates the model intercept (the model average when everything is kept at the reference category). The estimates have been transformed to match the outcome scale, for ease of interpretation.

**Figure 3 ijerph-18-04517-f003:**
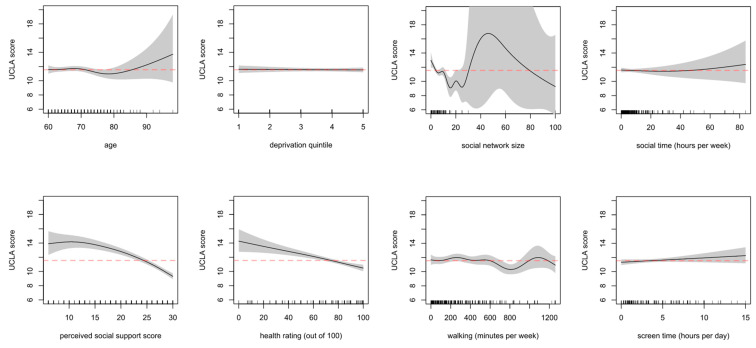
Associations with loneliness (UCLA) for continuous and ordinal variables on the outcome scale, from top left to bottom right: age, deprivation quintile, social network size, social time. From bottom left to bottom right: perceived social support (BPSSQ score), health rating, self-reported walking, and screen time. The dashed line indicates the model intercept (the model average when all other covariates are kept at their average/reference level).

**Figure 4 ijerph-18-04517-f004:**
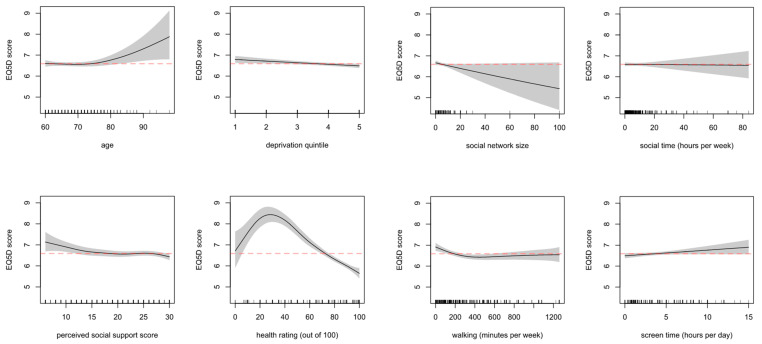
Associations with wellbeing (EQ5D-3L) for continuous and ordinal variables on the outcome scale, from top left to bottom right: age, deprivation quintile, social network size, social time. From bottom left to bottom right: perceived social support (BPSSQ score), health rating, self-reported walking, and screen timeThe dashed line indicates the model intercept (the model average when all other covariates are kept at their average/reference level).

**Table 1 ijerph-18-04517-t001:** Participant descriptive characteristics for participants aged 60+ years (total *n* = 1198).

Variable		*n*	Mean (SD)/*n* (%)
Sex	Female	1196	920 (77)
Age (years)		1198	67.3 (5.4)
Relationship	Single	1191	78 (7)
	Divorced/widowed		306 (26)
	In a Relationship		53 (4)
	Married/cohabiting		754 (63)
Health condition	Yes	1198	624 (52)
Carer	Yes	1198	149 (12)
Education	Did not complete	1168	72 (6)
	GCSE/O-levels ^1^		113 (10)
	Post-16 vocational course		40 (3)
	Highers/A-levels ^1^		121 (10)
	Undergraduate degree ^1^		545 (47)
	Postgraduate degree		277 (24)
SIMD deprivation quintile ^2^	1 (most deprived)	1094	89 (8)
	2		142 (13)
	3		229 (21)
	4		282 (26)
	5 (least deprived)		351 (32)
Urban/rural 3-fold classification ^2^	‘Rest of Scotland’ ^3^	1094	826 (69)
	Accessible rural		199 (17)
	Remote rural		69 (6)
Walking (min/week)		1193	336.2 (307.6)
Sleep category	Short sleeper	1187	520 (44)
	Ideal sleeper		616 (52)
	Long sleeper		51 (4)
Physical Activity category	Low active	1193	252 (21)
	Moderately active		426 (36)
	Highly active		515 (43)
Screen time (hr/day)		1193	3.7 (2.6)

^1^ Or equivalent. ^2^ SIMD values determined using valid postcodes (*n* = 131, 9% did not provide a valid Scotland postcode).^3^ Rest of Scotland’ includes Large Urban Areas, Other Urban Areas, Accessible Small Towns, and Remote Small Towns.

**Table 2 ijerph-18-04517-t002:** Participant loneliness, wellbeing, and social activity including social support.

Variable	Mean (SD)/*n* (%)
UCLA loneliness score	12.7 (4.70)
Social network size	5.5 (5.06)
Social contact (days per week)	5.4 (1.92)
Social time (hours per week)	7.0 (8.68)
Perceived social support (BPSSQ)	3.8 (1.04)
EQ5D-3L total score ^1^	6.7 (1.60)
Mobility	
*I have no problems in walking about*	919 (77)
*I have some problems in walking about*	278 (23)
*I am confined to bed*	1 (<1)
Self-care	
*I have no problems with self-care*	1103 (92)
*I have some problems washing or dressing myself*	92 (8)
*I am unable to wash or dress myself*	3 (<1)
Usual activities	
*I have no problems with performing my usual activities*	879 (73)
*I have some problems with performing my usual activities*	289 (24)
*I am unable to perform my usual activities*	30 (3)
Pain/discomfort	
*I have no pain or discomfort*	547 (46)
*I have moderate pain or discomfort*	598 (50)
*I have extreme pain or discomfort*	53 (4)
Anxiety/depression	
*I am not anxious or depressed*	646 (54)
*I am moderately anxious or depressed*	510 (43)
*I am extremely anxious or depressed*	42 (4)
Current health rating (out of 100) ^2^	72.5 (19.91)

^1^ A higher EQ5D-3L score = poorer health; ^2^ Both mean ± SD and median (IQR) shown (in that order).

**Table 3 ijerph-18-04517-t003:** Perceived changes in participants’ loneliness, wellbeing, and social activity including social support, and physical activity since pre-social distancing.

Variable	*n*		*n* (%)	
		Less	Same	More
Loneliness ^1^	1198	51 (4)	495 (41)	652 (54)
Health rating ^2^	1198	357 (30)	762 (64)	79 (7)
Anxiety/Depression ^1,2^	1198	48 (4)	733 (61)	417 (35)
Social network size	1194	690 (58)	373 (31)	131 (11)
Social contact frequency	1177	823 (69)	247 (21)	107 (9)
Social contact time	1176	835 (70)	243 (20)	98 (8)
Social support quality	1175	252 (21)	803 (67)	120 (10)
Giving social support	1174	227 (19)	516 (43)	431 (36)
Perceived Social support	1198	222 (19)	897 (75)	79 (7)
Walking	1198	572 (40)	480 (34)	377 (26)
Screen time	1198	42 (4)	416 (35)	740 (62)
Sleep volume	1198	433 (30)	915 (64)	81 (6)

^1^ For these, “more” indicates a negative outcome. ^2^ Phrased as worse, same, better, rather than less, same, more.

**Table 4 ijerph-18-04517-t004:** Unadjusted Pearson’s correlations between loneliness, social variables, and wellbeing.

	2.	3.	4.	5.	6.
1. Loneliness	−0.44 **	−0.17 *	−0.22 **	0.34 **	−0.31 **
2. Perceived Social support		0.40 **	0.27 **	−0.32 **	0.20 **
3. Social network size			0.13 *	−0.16 *	0.02
4. Social contact time				−0.08	0.12
5. EQ5D-3L total score					−0.55 **
6. Health rating					-

* *p* < 0.05, ** *p* < 0.01.

## Data Availability

The data presented in this study are available on request from the corresponding author. The data are not publicly available yet due to continued analysis but will be archived in Worktribe at the University of Stirling and. made publically available. following final publication acceptance or 12 months following the end of funding which was October 2020.

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
