# Peer review of "Loneliness, Wellbeing, and Social Activity in Scottish Older Adults Resulting from Social Distancing during the COVID-19 Pandemic"

_ijerph, 2021, doi:10.3390/ijerph18094517_

Round 1

Reviewer 1 Report

With great interest I read the manuscript on the impact of social distancing on loneliness, wellbeing, and social activity, including social support, in Scottish older adults. I think this is a great and well-performed study. I think the paper tries to do and include too much if anything.

My main concern is the inclusion of the smaller <60’s. This seems to be a fairly different, and heterogeneous group. This group is excluded from most of the analyses and shows unique themes within the qualitative study. For the focus on older adults, I would omit this group from the paper (and written up, maybe adding more descriptives, as a separate paper), or really incorporated it in the current paper. Potentially even comparing groups, although they differ in more than just age.

This paper in addition includes information that is not really linked to the research question or referred to after discussing it. For example, health and health behaviours. The paper is high information density, I don’t see what this actually adds to answering the research question. To keep the paper a little more focused either remove this information or incorporated it so it serves a purpose.

There is a recent study published on mental health in the Scottish population (https://doi.org/10.1007/s12529-021-09967-z), that shows that older adults are doing better in terms of anxiety and depression, the effect between age and psychological distress is moderated by loneliness, and illness representations for anxiety, which actually shows that the effect of loneliness on anxiety and depression is more pronounced for younger people. Potentially, the experienced loneliness is more in contrast with the current situation for young people and therefore affects them more. There is data on differences between experiences now on before the pandemic, so the current manuscript could answer this question. Are the findings of this manuscript consistent with this study, or how would this paper’s findings align with those findings?

An IQR for age would help with interpreting the age effects, as it stands it seems there are very few people over 80? The results on social network size seem really unreliable, would categorizing these continuous data give more useful information? Since, gender affects loneliness are interactions between gender and age, social support, and general health assessed?

The data was collected over a longer period, it would be really interesting to see if timing within the pandemic, phase of lockdown, affected experienced mental health, loneliness, and social support.   

Finally, the link between the quantitative and qualitative findings is not completely clear, how does frustration with the NHS link to loneliness and wellbeing for example? I think these parts could be written more aligned so the qualitative findings give some more depth and context to the findings, no they read more like 2 separate studies.  

Reviewer 2 Report

see attached file

Reviewer 3 Report

The paper by Tomaz et al. entitled "Loneliness, wellbeing, and social activity in Scottish older adults resulting from social distancing during the Covid19 pandemic" the impact of social distancing during the Covid19 pandemic on loneliness, wellbeing, and social activity, including social support, in Scottish older adults. In general, I find the study interesting and its results useful for the scientific community and for investigation. I am also pleased to point out that I greatly appreciated by the authors the self-criticism, which highlights the limitations of the study, but I believe that it is not enough to admit them and say "Future studies ....": there are some aspects that should be seriously revised.

Major

- please adapt the summary to clearly state the objectives so that it is well rounded (adapt the conclusion to fit the abstract).

- It is not clear what use they make of the information on physical activity, it seems that they want to exploit it in another article, as is the case with the sleep quality variable, but here they want to take advantage of including information, although only a little, and not all that is understood to have been collected. It is recommended that they make the decision to include or not to include or to do it in a more coherent way.

- Reconsider. If your article focuses on information from the over-60s, why complicate it with information from the under-60s, which you then fail to analyse or address. Perhaps you can leave that information for another article, or at the very least, reconsider not complicating or making it difficult to read. It does not provide information and it does complicate the reader.

- Please try to be clearer in the presentation of the information, to give more scientific basis to your work in the introduction, and above all, in the discussion of your results.

- Consider transferring information to supplementary material, e.g. transcripts of qualitative responses or figures 3 or 4. The suggestion is to try to make the article more accessible and streamlined for the reader, as such a density of inconclusive information does not encourage reading your work.

Overall a good and relevant study. Please improve the presentation. Having all aims clearly stated in the beginning and then again in the conclusion makes it much easier to read.

Minor

- The introduction lacks a broader theoretical basis. There is a lot of literature on the variables under study that the authors have omitted and I think it is important that they take the time to provide a basis for their work, so that later in the discussion they can discuss their results with those of other authors who have worked on this topics.

- L39. It talks about the objective of the study, but I think they should do it at the end of the introduction and not in the middle of it.

- L75. It includes a study of South Korea, but does not include studies in the European context, which I think is more appropriate and there is a large literature on the subject.

- L84 onwards. There is no data on physical activity information it seems that the authors consider working on physical activity elsewhere so it is recommended to take this into account throughout the article and not just omit it when discussing the results. Sometimes they include it, they say it, but they do not make it clear whether they want to expose it or not.

- L115.” No formal hypotheses about the direction and magnitude of perceived changes in psychosocial factors were formulated….”  I disagree. Why not?   But, then if hypotheses are stated. Please explain, because it is important to consider this in the discussion.

- L118-125 Not applicable in the article. Please focus your work on the variables under study, the context is relevant but not at this point in the introduction, where you should end with the objectives and hypotheses of the study.

- L174-175 “Five open-ended questions were used to explore any strategies adopted to maintain both physical (reported elsewhere) and social activity”. ¿Why? ¿Where?

- L184 “missing”. Please include the number of missing participants as indicated by the variable, social support. The same in L247.

- L 222. “seven days).” delete brackets

-L272-274. Where? They are doing another article on physical activity. Why do they say this one if they are talking about physical activity?

-L388.  As expected, these individuals were older (mean age = 81.8 versus 64.4 years). I don't understand the figure for the average age of the over-60s.  Where does 81.8 come from?

-L436. “3.6. Associations between loneliness, wellbeing and social activity including social support”.  In objective 3, they propose to explore the associations between psychosocial variables and physical activity during distancing, but I do not see in any section the work related to the achievement of this objective.  Please clearly state the results following the 5 specific objectives stated in the introduction.

-L520. “3.8. Strategies to maintain social contact”.  Consider summarising and including more content in supplementary material.

-L551 y 553. It is recommended to use the same format to talk about numbers or letters.

-L732…. Discusion

In general, I think it would be appropriate to review the discussion. Revise, not expand. I think it focuses more on presenting the results than on discussing them. I consider it relevant to provide a greater scientific basis for the discussion of the results obtained with those of other authors.

- References

Please check the references

- some of them are incomplete, others end with ; e.g. 2, 31....

- others do not include the doi, among others, e.g. 19, 28....

Author Response

Note that many of the comments from reviewer 3 overlapped with those of Reviewer 1 (mostly) and Reviewer 2. It may be helpful for Reviewer 3 to refer to those documents. 

Round 2

Reviewer 3 Report

I congratulate the authors on the improvement of the manuscript.